# Extracellular Polysaccharide Receptor and Receptor-Binding Proteins of the *Rhodobacter capsulatus* Bacteriophage-like Gene Transfer Agent RcGTA

**DOI:** 10.3390/genes14051124

**Published:** 2023-05-22

**Authors:** Nawshin T. B. Alim, Sonja Koppenhöfer, Andrew S. Lang, J. Thomas Beatty

**Affiliations:** 1Department of Microbiology & Immunology, University of British Columbia, Vancouver, BC V6T 1Z4, Canada; ntbalim@student.ubc.ca; 2Department of Biology, Memorial University of Newfoundland, St. John’s, NL A1C 5S7, Canada; skoppenhofer@mun.ca (S.K.); aslang@mun.ca (A.S.L.)

**Keywords:** *Rhodobacter*, gene transfer agent, RcGTA, horizontal gene transfer, phage, virus, head spike, tail fiber

## Abstract

A variety of prokaryotes produce a bacteriophage-like gene transfer agent (GTA), and the alphaproteobacterial *Rhodobacter capsulatus* RcGTA is a model GTA. Some environmental isolates of *R. capsulatus* lack the ability to acquire genes transferred by the RcGTA (recipient capability). In this work, we investigated the reason why *R. capsulatus* strain 37b4 lacks recipient capability. The RcGTA head spike fiber and tail fiber proteins have been proposed to bind extracellular oligosaccharide receptors, and strain 37b4 lacks a capsular polysaccharide (CPS). The reason why strain 37b4 lacks a CPS was unknown, as was whether the provision of a CPS to 37b4 would result in recipient capability. To address these questions, we sequenced and annotated the strain 37b4 genome and used BLAST interrogations of this genome sequence to search for homologs of genes known to be needed for *R. capsulatus* recipient capability. We also created a cosmid-borne genome library from a wild-type strain, mobilized the library into 37b4, and used the cosmid-complemented strain 37b4 to identify genes needed for a gain of function, allowing for the acquisition of RcGTA-borne genes. The relative presence of CPS around a wild-type strain, 37b4, and cosmid-complemented 37b4 cells was visualized using light microscopy of stained cells. Fluorescently tagged head spike fiber and tail fiber proteins of the RcGTA particle were created and used to measure the relative binding to wild-type and 37b4 cells. We found that strain 37b4 lacks recipient capability because of an inability to bind RcGTA; the reason it is incapable of binding is that it lacks CPS, and the absence of CPS is due to the absence of genes previously shown to be needed for CPS production in another strain. In addition to the head spike fiber, we found that the tail fiber protein also binds to the CPS.

## 1. Introduction

In prokaryotes, the term gene transfer agent (GTA) refers to bacteriophage-like elements that carry out horizontal gene transfer (HGT). HGT is important for genome evolution because the majority of genes found in bacterial genomes have undergone HGT at some point in their evolution [1,2]. Phage-mediated HGT (transduction) spreads antibiotic resistance and virulence genes in pathogens [2,3]. Up to 10^9^/mL of phage-like particles, including GTAs, are found in aquatic environments [4,5,6], GTAs have been found in archaea, spirochetes, α- and deltaproteobacteria, and a GTA was found to transfer antibiotic resistance in an aquatic microcosm [4,7,8]. Therefore, it is important to learn more about GTAs to understand their contributions to HGT evolution and to exploit their activities in translational applications. The GTA of the alphaproteobacterium *Rhodobacter capsulatus* (RcGTA) is a model system that may reveal basic insights into GTAs in general.

Recent cryo-EM work (Figure 1) showed structural similarity between RcGTA and tailed DNA phages [9] at a much higher resolution than was available from previous electron microscopy images [10,11]. Despite the structural similarity, in contrast to phages, GTAs solely package pieces of the cell genome for HGT. Production of RcGTA kills the producing cell, but processes have evolved that limit production to a small percentage of cells, <3% [12,13]. The rare induction of RcGTA gene expression is not due to a genetic change but instead is a stochastic process leading to a bistable gene expression pattern within a clonal population, repressed by a homolog of secreted Ca^2+^-binding repeat (hemolysin- and RTX-like) proteins [14]. It was also found that the remaining >97% of cells in a culture enter an alternative pathway and develop the ability to actively import genes carried by RcGTA. Quorum sensing regulates the biosynthesis of a RcGTA receptor, the capsule polysaccharide (CPS), produced by enzymes encoded by genes that are located adjacent to and co-regulated with the genes encoding RcGTA head spike proteins that interact with the CPS during initial RcGTA cell contact [11,15]. Additionally, uptake and incorporation of RcGTA-delivered DNA into the genome of recipients are dependent upon a phosphorelay-regulated import system that includes homologs of proteins needed for competence (natural transformation) in other species [16,17]. This population differentiation, with highly coordinated regulation of RcGTA production in a few cells and the ability to receive DNA from the RcGTAs in most cells, highlights the extent to which RcGTA is integrated into *R. capsulatus* physiology. This integration supports the notion that the production of RcGTA has been maintained through evolutionary time because it is beneficial to the bacterial species [18,19,20].

Although the ability to acquire RcGTA-borne genes is generally exhibited by most strains of *R. capsulatus*, some strains lack recipient capability [9]. The cryo-EM structural analysis of RcGTA revealed that the CPS-binding head spikes consist of eleven pentamers of a base protein attached to the capsid and a flexible head spike fiber bound to the base protein pentamer. Head-distal proteins, such as the tail fibers, contain predicted oligosaccharide-binding domains [10]. These domains have sequence and structural homology to corresponding domains of *Pseudomonas* phages and tail fibers from R-pyocins, which are known to bind to the core polysaccharide of lipopolysaccharides (LPS) [11,12]. Bardy et al. [10] presented a model in which the head spikes enable RcGTA to bind to the CPS of the cell, followed by movement of RcGTA through the CPS and a switch from random orientation to a posture with the baseplate facing the cell, for the tail fibers to bind to another ligand on the cell outer membrane. The head spike interaction with CPS is not an obligatory step for gene transfer because cells lacking CPS are able to receive genes from RcGTA, and RcGTA lacking the head spikes can donate genes to cells, albeit at a low frequency [13,14,15]. The interaction of the tail fiber appears to be essential, however, for the transfer of DNA into the periplasm of the cell [16].

The flexibility of the RcGTA head spike fiber protein interfered with high-resolution structure determination, apart from the N-terminal ten amino acids that interact with the pentamer of spike base protein [10], and so it is absent from the image shown in Figure 1. However, we obtained an AlphaFold [17,18] structure prediction that was used to design a fluorescent protein-tagged head spike fiber.

The RcGTA tail fiber protein consists of several domains, including a C-terminal region that appeared to be flexible and did not allow for structure determination [10], and therefore only the N-terminal regions are shown in Figure 1. However, an AlphaFold-derived [17,18] structure prediction was obtained and used to design a fluorescent protein-tagged tail fiber.

We investigated the reason why the environmental isolate *R. capsulatus* strain 37b4 lacks recipient capability. Strain 37b4 lacks a CPS, and recipient capability is undetectable, whereas CPS biosynthetic mutants of a wild-type (WT) strain retain about 10% recipient capability [14]. The difference between the recipient capability of strain 37b4 and CPS-less mutants led us to speculate that 37b4 lacks two receptors (the CPS and perhaps another receptor), whereas CPS-less mutants lack only the CPS and retain a second receptor. Furthermore, the genetic basis for why strain 37b4 lacks a CPS was unknown.

To address these questions, we sequenced and annotated the strain 37b4 genome and used BLAST interrogations of this genome sequence to search for genes known to be needed for *R. capsulatus* CPS production, which might be absent from the 37b4 genome. We also created a cosmid-borne genome library from a WT strain, mobilized the library into 37b4, and used the cosmid-complemented 37b4 strain to select for a gain of the function allowing for the acquisition of RcGTA-borne genes.

The binding of RcGTA head spike fiber and tail fiber proteins to cells containing or lacking CPS was evaluated directly by using fluorescent protein-tagged head spike and tail fiber proteins. We conclude that the absence of CPS is a reason for the absence of recipient capability in strain 37b4 and that the RcGTA tail fiber, as well as the head spike fiber protein, binds to the CPS.

## 2. Materials and Methods

### 2.1. Strains, Growth Conditions, and Plasmids

The *Escherichia coli* strain DH5α was used for cloning, and strain S17-1 [19] was used for the conjugation of plasmids into *R. capsulatus*. *E. coli* strain HB101(pRK2013) [20] was used as the helper strain in some conjugations. *E. coli* BL21 DE3 [22] was used for isopropyl β-D-1-thiogalactopyranoside (IPTG) induction of recombinant protein production for purification. *E. coli* strains were grown at 37 °C in a Luria–Bertani (LB) medium [23] supplemented with the appropriate antibiotics at the following concentrations in µg/mL: tetracycline, 10; kanamycin sulfate, 50.

The *R. capsulatus* WT strain B10 [24] was the source of DNA for the construction of the cosmid-borne WT genomic library. The WT strain SB1003, a rifampicin-resistant derivative of B10 [25], the non-recipient environmental isolate 37b4 [26], the RcGTA overproducer strain DE442 [27], the kanamycin-resistant overproducer strain DE2539 [16], and the quorum-sensing mutant *∆gtaI*, which lacks a capsule [14], have been described. The WT B10 genomic DNA-complemented strain 37b4(pCPS1) was isolated in the work described here. All *R. capsulatus* strains were grown in a defined RCV medium [28] or complex medium YPS [9] at 30 °C. Antibiotics used for selection when appropriate were in the following concentrations (µg/mL): tetracycline 0.5; kanamycin 10.

The culture growth phase (turbidity) was monitored in a Klett–Summerson photometer (red filter #66); 100 Klett units represent approximately 4 × 10^8^  *R. capsulatus* colony-forming units per ml (cfu/mL).

Cosmid pLAFR1 [29] and plasmid pET28a(+) (Twist Bioscience, South San Francisco, CA, USA) were used in library construction and as recombinant protein overexpression vectors, respectively. The cosmid pCPS1 was isolated in the work described here.

### 2.2. WT R. capsulatus Genome Library Construction and Recipient Capability Complementation of Strain 37b4

The construction of the genome library in cosmid pLAFR1 and conjugation from *E. coli* into *R. capsulatus* 37b4 were carried out as described [30], with selection for the cosmid-encoded tetracycline resistance. Tetracycline-resistant colonies of 37b4 were resuspended to a turbidity of 140 Klett units and grown into the stationary phase, after which they were used in a gene transfer experiment with RcGTA obtained from the kanamycin-resistant strain DE2539, as described [31]. The cosmids in kanamycin and tetracycline doubly-resistant cells of colonies that arose were purified and transformed into *E. coli* DH5α, and subsequently used for Sanger sequencing with primers that bind to either side of the *Eco*RI site into which the genomic DNA sequences had been inserted to determine the region of the WT strain B10 genome that was present in cosmid pCPS1.

### 2.3. RcGTA Recipient Capability and Adsorption Assays

Strain recipient capabilities were quantified using the gene transfer assay as described [31], with RcGTA obtained from the kanamycin-resistant strain DE2539, selecting for kanamycin resistance.

The RcGTA adsorption assay was done as described [14], with RcGTA obtained from strain DE2539 and SB1003 cells as the recipient.

### 2.4. Capsule Stain

Capsules were negatively stained using Anthony’s capsule stain [32]. In brief, 100 µL of stationary phase *R. capsulatus* cultures were harvested by centrifugation and suspended in 100 µL of 5% skim milk (skim milk diluted *v*/*v* with deionized water). Twenty microliters of each sample were spread onto a clean microscope slide and allowed to air-dry at room temperature. Dried samples were then stained with 1% crystal violet for 2 min and washed gently and thoroughly with 20% CuSO_4_. The sample was immediately covered with a coverslip, pressed down with paper tissue to force out and wick off excess liquid, and the coverslip was sealed with clear nail polish. The samples were subsequently examined using oil-immersion phase contrast microscopy at 100× magnification.

### 2.5. Genome Assembly and Alignment Analysis

Short-read paired-end sequencing was performed on an Illumina MiSeq platform at the Integrated Microbiome Resource facility (Halifax, NS, Canada). Auxiliary sequences were trimmed off using Trimmomatic v. 0.39 [33] with the following specifications: sliding window size: 4, average window quality: 20; minimum trailing quality: 20; head crop length: 9; minimum length of reads kept: 100 bp.

Long-read sequencing was performed using the Oxford Nanopore Technologies (ONT) MinION system [34]. Long-read sequences were basecalled using the ONT Guppy basecaller (v. 4.5.2) with the –recursive and --calib_detect options and the following specifications: configuration file DNA_r9.4.1_450bps_hac.cfg; device cuda:all:100%; minimum quality score 8. Using Filtlong v. 0.2.0 (github.com/rrwick/Filtlong), the concatenated sequences were filtered by quality to keep reads with a minimum length of 400 bp (‘--min_length 400’) and throw out the worst 10% of reads (‘--keep_percent 90’).

De novo hybrid sequence assembly of strain 37b4 was performed using Unicycler v0.5.0 [35] with hybrid assembly options and default bridging mode. Prokka v. 1.14.5 [36] was used for the annotation of the assembled 37b4 genome with the SB1003 genome as the reference. Locus tags referred to in Appendix A are derived from this reference-based annotation. Annotation of the published genome was performed using NCBI’s Prokaryotic Genome Annotation Pipeline (v. 6.4).

The assembled 37b4 genome was globally aligned against the WT SB1003 genome (accession number CP001312.1) using progressiveMauve (https://darlinglab.org/mauve/mauve.html (accessed on 22 December 2022)) with default settings [37].

### 2.6. BLAST Analysis

The blast+ suite v2.12.0 was used for this set of analyses. A custom protein database was created from the assembled 37b4 Prokka v. 1.14.5 annotation [36]. Protein sequences encoded in WT SB1003 genes were used to query the 37b4 database using BLASTp. Top hits (lowest E-value) from this search were used in a reciprocal BLASTp analysis of the SB1003 WT reference genome to evaluate the matches initially obtained in the BLASTp of the 37b4 genome with SB1003 sequences.

### 2.7. Creation of Overexpression Constructs and Recombinant Fluorescent Protein Purification

Two overexpression pET28a(+) vectors were designed with inserts encoding a 6 His-tag at the N terminus of the mNeonGreen fluorescent protein, fused to the N terminus of either the *R. capsulatus* gene *rcc01080* (RcGTA head spike fiber) or *rcc00171* (RcGTA tail fiber), with the linker sequence SGLRSPPVAT [38] intervening between the C-terminal K residue of the fluorescent protein and the N-terminal M of the tail fiber, or the A residue #11 of the head spike fiber protein. The vectors were purchased from Twist Bioscience (South San Francisco, CA, USA) and designated as pET28a::RcHS (tagged head spike fiber) or pET28a::RcTF (tagged tail fiber).

Induction of recombinant protein overexpression in *E. coli* BL21 DE3(pET28a::RcHS or pET28a::RcTF) was performed using IPTG as described by Hynes et al. (2016). Induced cells were harvested by centrifugation, and cell pellets were resuspended in lysis buffer (50 mM NaH_2_PO_4_, 300 mM NaCl; pH 7.8). Suspended cell pellets were lysed by passage through a French press cell (Aminco) at 1250 psi. The resultant lysate was centrifuged at 34,500 rcf for 15 min to produce a cleared lysate, which was incubated with Ni-NTA agarose beads (Qiagen) overnight. The protein purification was performed as described by Hynes et al. (2016) with the following modifications: washing and elution steps were carried out with 10 column volumes of respective buffer; concentration of purified protein was determined by measuring optical density at 517 nm; buffer exchange and concentration of the proteins (to OD_517nm_ ≥ 8) were performed using a 40 kDa cutoff-ultrafiltration device (Amicon Ultra-0.5) and the storage buffer (20 mM NaH_2_PO_4_, 500 mM NaCl, pH 6.8); purified proteins were stored at −80 °C.

### 2.8. Fluorescence-Based Binding Assay

*R. capsulatus* strains were grown photoheterotrophically to the stationary phase (≥400 Klett units) to ensure that encapsulated strains would have the capsule as a result of quorum-sensing induction [14]. One volume of cell culture was fixed by adding 7% (*v*/*v* final concentration) formaldehyde and incubation for 20 min at room temperature with gentle agitation and washing with modified G-buffer (10 mM Tris-HCL, 1 mM NaCl, 1 mM CaCl_2_, 1 mM MgCl_2,_ pH 6.8). The cell pellets were suspended in five volumes of modified G-buffer, recombinant head-spike fiber or tail fiber protein was added to the cell suspension to a final concentration of 0.005 to 0.01 µg/µL (thereby ensuring cells were in excess) and incubated for 20 min at room temperature with gentle agitation. The cells were pelleted by centrifugation, and the fluorescence at 517 nm of the resultant supernatant was measured using a StellarNet (Tampa, FL, USA) spectrofluorometer equipped with a BLACK-Comet spectrometer, 502 nm Green LED excitation light (StellarNet SL1-LED Part # 67-1755), and data viewed using StellarNet SpectraWiz software v. 5.33. Cell-free control experiments of the respective recombinant proteins in a modified G-buffer were performed to account for protein aggregation. Three biological replicates were performed for each strain.

## 3. Results

### 3.1. Comparison of WT SB1003 and 37b4 Genomes, with a Focus on Genes Needed for Recipient Capability

The SB1003 genome consists of a chromosome of about 3.7 Mb in length and an approximately 133 kb plasmid [https://www.ncbi.nlm.nih.gov/Taxonomy/Browser/wwwtax.cgi?id=272942 (accessed on 22 December 2022)]. Strain 37b4 lacks a plasmid, and the chromosome is about 3.9 kb in length. The average nucleotide identity (ANI) of the two chromosomes was calculated using FastANI v. 1.33 [39] as 93.7%, which is slightly lower than the operational cutoff between species of 95%, although there are exceptions to this loose definition [40]. As shown in the Mauve alignment [37] of Figure 2, the SB1003 and 37b4 chromosomes are predominantly composed of locally colinear blocks of nucleotide sequence similarity (shown by the colors with lines connecting the homologous sequences in the two chromosome representations). However, there are several regions lacking homologous sequences, as shown by the gaps between colored blocks, some translocations, and inversions, particularly in the region within the 1.1 to 2.7 Mb coordinates (Figure 2).

Multiple *R. capsulatus* genes have been identified as being required for recipient capability. The proteins encoded by these genes function in processes such as the production of CPS needed for binding of RcGTA, transport of DNA across the cytoplasmic membrane, and recombination of the incoming DNA into the recipient cell’s genome [14,41,42]. Notably, an SB1003 CPS biosynthesis gene cluster (*rcc01081*-*01086*, located at approximately 1,150,000 to 1,160,000 nucleotides from the *dnaA* start codon) is located in a region lacking homology to 37b4 in the Mauve alignment of Figure 2.

We used BLASTp to search the strain 37b4 genome for homologs of SB1003 genes previously implicated in recipient capability and evaluated top hits in reciprocal BLASTp searches [43,44]. As shown in Table 1, strain 37b4 encodes homologs of some of the proteins needed for recipient capability, including processes such as the transport of DNA across the cytoplasmic membrane and DNA recombination. However, the entire CPS biosynthesis gene cluster noted above appears to be absent from 37b4, consistent with alignment in Figure 2. Other putative genes are present on cosmid pCPS1 and absent from 37b4 (see Appendix A), but only *rcc01081* to *rcc1086* are known to be required for recipient capability.

### 3.2. Genes Conferring RcGTA Binding, Recipient Capability, and the Presence of a Capsule on Strain 37b4

We created a gene library in the broad-host-range cosmid pLAFR1 [29] using DNA from the WT strain B10. The cosmid-borne library was conjugated en masse into strain 37b4, and cosmid-containing cells were incubated with RcGTA produced by an SB1003 derivative in which a kanamycin-resistance cartridge had been inserted into the CDS *rcc02539* [16]. Cells were plated on an agar medium containing kanamycin to select for a potential gain of recipient capability. Kanamycin-resistant colonies were obtained, and the cosmids present were evaluated by purification and transformation into *E. coli*, conjugation into strain 37b4, and testing to confirm that the recipient capability trait was encoded by the cosmid.

One such cosmid was found, and the *R. capsulatus* sequence in this cosmid (designated pCPS1) was identified by Sanger sequencing from each end of the insert, revealing an approximately 22 kb sequence extending from within *rcc01067* to within *rcc01088* (Figure 3). This region contains the proposed CPS biosynthetic genes *rcc01081* to *rcc01086* that appeared to be absent from 37b4 in the Figure 2 alignment and in the BLASTp analysis summarized in Table 1.

The recipient capability of the pCPS1 cosmid-complemented 37b4 strain relative to the WT strain SB1003 was quantitatively evaluated by mixing equal numbers of cells with equal numbers of RcGTAs and selecting for the transfer of a kanamycin-resistance marker. As shown in Figure 4, it was found that the relative recipient capability (expressed as relative numbers of kanamycin-resistant colonies resulting from gene transfer) of strain 37b4 was zero, whereas that of 37b4(pCPS1) was 11.5%, relative to strain SB1003. Therefore, although the cosmid pCPS1 provided strain 37b4 with genes essential for recipient capability, the frequency of gene transfer was less than that of the WT strain SB1003.

The difference in recipient capability (the number of kanamycin-resistant colonies obtained) between strains SB1003 and 37b4(pCPS1) may be due in part to the lower DNA sequence identity that we identified between the regions flanking the kanamycin-resistance cartridge in the RcGTA donor strain DE2539 and the corresponding regions in the chromosome of the 37b4 recipient strain. Strain DE2539 [45] was constructed by the insertion of a kanamycin-resistance cartridge into the *Sma*I site of the *rcc02539* gene of strain DE442. The DE442 genome has been sequenced [27], and looking specifically at the 200 bp on either side of the kanamycin-resistance cartridge inserted into the *Sma*I site in the *rcc02539* sequence of SB1003 and 37b4, we found 16 mismatches per 200 bp on the 5′ side and 13 on the 3′ side (6.5 and 7%), shown in Table 2. In contrast, the SB1003 strain had 0 and 1% mismatches, respectively. Extending the length of comparison to 400 bp on either side of where the kanamycin-resistance cartridge is inserted had little effect on these numbers: we found 33 mismatches per 400 bp on the 5′ side and 25 on the 3′ side (8.2 and 6.2%) for strain 37b4, whereas SB1003 had 0 and 0.5% mismatches, respectively.

The binding of RcGTA to cells was quantified by mixing equal numbers of cells with a constant amount of RcGTA particles, removal of cell-bound RcGTA by centrifugation, and assay of the relative amount of non-bound RcGTA by measurement of the frequency of transfer of the kanamycin-resistance marker. Figure 5 shows that, on average, the percentage of cell-free RcGTA after incubation with strain 37b4 was 72% of the no cells control, whereas the WT strain SB1003 value was 5%. Therefore, the WT SB1003 bound 3.4-fold greater numbers of RcGTAs than the 37b4 cells. This is consistent with the absence of CPS from strain 37b4 that is present in SB1003 and that the CPS is needed for maximal binding of RcGTA. Strain 37b4(pCPS1) bound RcGTA better than 37b4, but it only reached 50% in comparison to the 95% value for SB1003. It appears that there are RcGTA receptors present on SB1003 that are absent from 37b4(pCPS1) or that the CPS is not produced well. We address this issue in the Discussion section.

To directly demonstrate whether a capsule is present in strains SB1003, 37b4, and 37b4(pCPS1), we first used a cell centrifugation assay that had been previously employed [14,15]. In this assay, equal numbers of cells are centrifuged, and the absence of a capsule is indicated by a relatively small and tight cell pellet compared to an encapsulated strain such as SB1003. It was found that strain 37b4(pCPS1) exhibited a relatively large, diffuse, and mucoid cell pellet compared to the parental strain 37b4 and similar to that of SB1003 (Figure 6).

We also employed Anthony’s stain [32] to visualize the presence or absence of a capsule in microscopy. Figure 7 shows a bright zone due to the capsule around cells of strain SB1003, the absence of such a zone around cells of 37b4, and the presence of a similar, slightly less bright zone around cells of 37b4(pCPS1).

Taken together, these results confirm the interpretation that strain 37b4 is unable to receive RcGTA-borne genes because it is unable to bind RcGTA due to the absence of CPS resulting from the absence of CPS biosynthetic genes *rcc01081* to *rcc01086*, which are present on the cosmid pCPS1 (Figure 3).

### 3.3. Binding of Fluorescent Protein-Tagged Tail Fiber and Head Spike Proteins to the WT Strain SB1003, the Quorum-Sensing Mutant ΔgtaI (a gtaI Knockout Lacking a Capsule), 37b4, and 37b4(pCPS1)

A model has been proposed for RcGTA binding to cells in which the RcGTA head spike fibers interact with the CPS, followed by RcGTA migration to the cell surface with subsequent binding of the tail fibers to a cell surface oligosaccharide receptor [10]. A corollary of this model is that the head spike and tail fiber proteins might interact differentially with extracellular receptors, with the head spike protein binding strongly to the CPS and weakly to a cell surface receptor and the tail fiber protein binding weakly to the CPS and strongly to a cell surface receptor. The head spike fiber protein and much of the tail fiber protein appear to be flexible, adopting multiple conformations and therefore were not well-resolved in the cryo-EM structures presented previously [10]. Therefore, as a first step in testing this model, we used the AlphaFold program [17,18] to predict the head spike and tail fiber structures.

Figure 8 shows the AlphaFold model of the tail fiber, which previously had been solved by cryo-EM to a resolution of 6.8 to 13.9 Å for the tail-proximal N-terminal domains called the rod and the knob [10]. The less well-resolved C-terminal domain (the “foot”) is homologous to sugar-binding structures of pyocin fibers, and the AlphaFold model predicts β strands organized similarly to those of pyocins. Because the N terminus is predicted to be far from the C-terminal foot, thought to bind to a sugar receptor, we designed a synthetic gene encoding a fluorescent protein attached to the N terminus, as shown in Appendix A.

The fluorescent protein-tagged tail fiber was used in experiments to evaluate binding to cells and whether the capsule affects binding. As shown in Figure 9, 100% of this tagged tail fiber protein bound to the WT strain SB1003, whereas the *ΔgtaI* mutant (which lacks a capsule and is a control in this experiment) bound about 20% of the protein. The 37b4 strain, which also lacks a capsule, bound about 55% of the tagged tail fiber protein, whereas the presence of the pCPS1 cosmid in 37b4 cells increased the binding to almost the level of SB1003, or to 9% of the unbound protein using the *ΔgtaI* mutant and 16% of the unbound protein using the 37b4 strain lacking the cosmid. Therefore, the presence of cosmid pCPS1 in strain 37b4 increased the binding of the tagged tail fiber protein about 6-fold. These results indicate that strain 37b4 binds a greater percentage of the tail fiber than the *ΔgtaI* mutant, and the presence of the CPS increases the binding of the tail fiber to cells.

2.The cryo-EM structure of the head spike fiber protein contained high-resolution data of only the N-terminal residues 2 to 10, which showed close interaction (3.5 to 4.9 Å) with residues of the spike base protein pentamer [10]. Therefore, the head spike fiber appears to be anchored by the N terminus, with the bulk of the protein available for interaction with a receptor previously proposed to be the CPS [15]. An AlphaFold model of the head spike fiber protein is given in Figure 10, showing an extended N terminus that interacts with the base protein and a quasi-two domain structure dominated by β strands and loops, as in lectins and other sugar-binding proteins. This AlphaFold structure probably represents only one computationally low-energy-predicted conformation of several structural states. Because the head spike fiber N terminus was found to interact with the base protein pentamer for attachment to the capsid [10], whereas the rest of the protein is thought to bind to a CPS sugar receptor, we designed a synthetic gene encoding a fluorescent protein attached to the N-terminal residue 11, as shown in Appendix A.

3.The fluorescent protein-tagged head spike fiber was used in experiments to evaluate binding to cells. As shown in Figure 11, 100% of this tagged protein bound to the WT strain SB1003 cells, whereas the *ΔgtaI* mutant (which lacks a capsule) bound about 4%. The 37b4 strain, which also lacks a capsule, bound about 45%, whereas the presence of the pCPS1 cosmid in 37b4 cells increased the binding to about 72%, about 1.6-fold more than the 37b4 strain lacking the cosmid. These results indicate that, as for the tail fiber, the presence of the CPS increases the binding of the head spike fiber. Furthermore, the 37b4 strain bound a percentage of the head spike fiber protein intermediate between the WT strain SB1003 and the cosmid-complemented strain 37b4(pCPS1).

## 4. Discussion

In this work, we investigated why the *R. capsulatus* strain 37b4 lacks RcGTA recipient capability. We used several approaches, ranging from genome sequence comparisons to measuring the binding of two putative receptor-binding proteins to cells. Our results show that an important reason for the inability of strain 37b4 to acquire RcGTA-borne genes is the absence of CPS, which in turn is because of the absence of a CPS biosynthetic gene cluster, designated *rcc01081* to *rcc01086* in the genome sequence of the WT strain SB1003.

The absence of homologs of the WT SB1003 CPS biosynthetic gene cluster was indicated by the Mauve alignment shown in Figure 2. This absence was confirmed by the BLASTp query of the 37b4 genome sequence, using the amino acid sequences of the proteins encoded by the genes *rcc01081* to *rcc01086* (Table 1). The presence of genes *rcc01081* to *rcc01086* on the cosmid pCPS1, which endowed strain 37b4 with recipient capability, provides additional support for this interpretation.

The absence of CPS in strain 37b4 was shown in a cell-pelleting centrifugation assay and directly by microscopy of cells using a capsule staining technique, as was the presence of CPS in 37b4(pCPS1) (Figure 7). The capsule around strain 37b4(pCPS1) appeared to be less pronounced than in the WT SB1003, and so there may be additional physiological factors affecting CPS development that are lacking in 37b4(pCPS1).

RcGTA recipient capability frequencies were lower in strain 37b4(pCPS1) than in the WT SB1003 strain (Figure 4). The difference in gene transfer frequencies between the WT strain vs. 37b4 and 37b4(pCPS1) may be in part because the region of the 37b4 chromosome homologous to the kanamycin resistance cartridge-disrupted *rcc02539* sequence that was transferred has more mismatches in strain 37b4 than SB1003 compared to strain DE442 (Table 2), the parental strain of DE2539 from which the RcGTA particles carrying the kanamycin resistance cartridge were obtained. Because RcGTA-mediated gene transfer requires homologous recombination, and homologous recombination frequencies decrease as two sequences diverge, such mismatches may contribute to the difference in recipient capability between the RcGTA-production strain DE442 vs. 37b4 and 37b4(pCPS1). A high frequency of recombination (i.e., in our case, transfer of kanamycin resistance) has been shown in *E. coli* to require at least 200 bp of identical sequences on both sides of the region being recombined, and recombination frequencies increase with longer regions of identity [46,47].

However, before DNA is injected into a potential recipient cell, binding of RcGTA to the cell is needed. It was previously shown that the head spike is needed for binding to the CPS [15] and that the addition of the tail fiber to cells blocks gene transfer, proposed to be due to binding to and blocking of a receptor [45]. The binding of RcGTA is thought to involve five proteins that were found to contain domains homologous to proteins that bind oligo- or polysaccharides: the head spike fiber, the distal tail, the hub, the megatron, and the tail fiber proteins. Because these proteins are present in multiple copies, the total number of predicted sugar-binding domains is 32 [10]. Therefore, the binding of RcGTA particles to cell surface receptors is likely to be complex and possibly involves interactions with multiple sugars.

We used two general methods to evaluate the binding of RcGTA to cells: (i) binding of RcGTA particles, detected by a decrease in gene transfer frequency after removal of cells (and bound RcGTA); (ii) binding of fluorescent protein-tagged recombinant head spike fiber or tail fiber proteins to cells, detected by a decrease in fluorescence after removal of cells (and bound fluorescent protein). All of these data clearly show that the strain 37b4 has weaker binding than the WT SB1003 and that 37b4(pCPS1) has stronger binding than 37b4 and the quorum sensing mutant *ΔgtaI*, which had previously been shown to lack a CPS and to be impaired in the binding of RcGTA [14].

The binding of RcGTA to the *ΔgtaI* mutant had previously been found to be greater by 20% compared to a no-cells control [14]; here, we found that binding to strain 37b4 was greater by 28% (Figure 5), and so 37b4 appears to bind RcGTA more strongly than the *ΔgtaI* mutant. Both the fluorescent protein-tagged tail fiber and head spike appeared to bind to strain 37b4 more strongly than to the *ΔgtaI* mutant (Figure 9 and Figure 11). The *ΔgtaI* mutant is deficient in quorum sensing, whereas 37b4 contains an intact *gtaI* gene encoding a homolog 98% identical to the SB1003 GtaI protein. In *R. capsulatus,* quorum sensing induces multiple physiological processes in addition to CPS biosynthesis, including RcGTA production and cell lysis, and expression of several genes encoding proteins involved with the import of RcGTA-borne genes into recipient cells [41,48]. We suggest that quorum sensing is needed for the production of one or more additional factors, present in strain 37b4 and absent from *ΔgtaI*, which are involved in the binding of these proteins to cells.

Furthermore, it was also found that the binding of RcGTA to strain 37b4(pCPS1) was approximately 50% of the binding to the WT SB1003 (Figure 5), whereas the binding of the tagged tail fiber and head spike fiber proteins was 93% and 72%, respectively, of the binding to SB1003 (Figure 9 and Figure 11). This lesser binding probably contributes to the decreased recipient capability of 37b4(pCPS1) relative to SB1003 noted above (Figure 4). We attribute the differences in binding of RcGTA to these two strains to differences in cell-surface receptors involved in multiple interactions between receptor-binding domains of the distal tail, the hub, and the megatron proteins. In contrast, the tail fiber and head spike fiber protein ligands on the surface of strain 37b4(pCPS1) appear to be more similar to those on the WT strain SB1003 because of the more similar binding percentages.

In the future, it will be interesting to employ high-resolution imaging of RcGTA and predicted receptor-binding proteins bound to their ligands, and biochemical characterization of ligands, to improve our understanding of this complex, multi-component suite of interactions that are involved in recognition of potential recipients in this gene transfer process.

## 5. Conclusions

The binding of RcGTA to cells for efficient delivery of DNA appears to involve several receptor-binding proteins that contain predicted sugar-binding domains, which may bind to different ligands as well as share ligands. We show here that ligand-sharing is the case for two RcGTA proteins located on opposite ends of the RcGTA particle, the head spike fiber and the tail fiber proteins. Although the CPS is an important component in the binding of these two proteins, they do not share detectable homology in a BLASTp alignment. Therefore, it is possible that the head spike fiber and the tail fiber protein bind to different components of the CPS. Additional receptors may include the lipopolysaccharide (LPS), which itself may contain an O-antigen polysaccharide, outer and inner core sugars, and the glucosamine disaccharide of lipid A [49].

## Figures and Tables

**Figure 1 genes-14-01124-f001:**
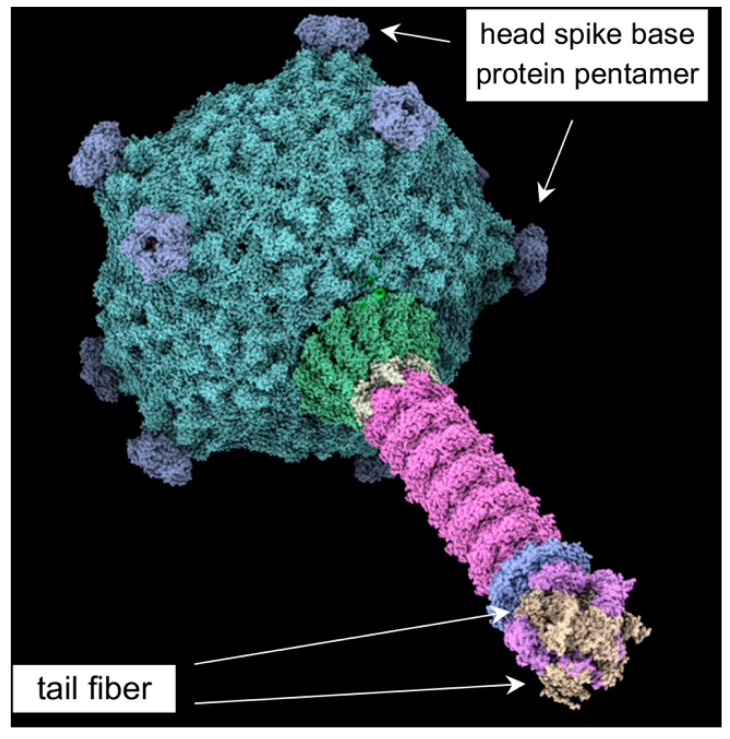
Cryo-EM structure of RcGTA (pdb 6tba). Image obtained using *ChimeraX* [21].

**Figure 2 genes-14-01124-f002:**
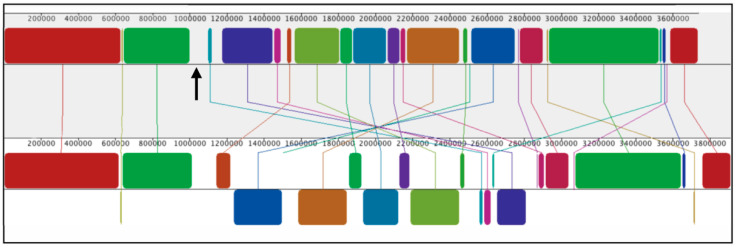
Graphical alignment of the WT strain SB1003 (top) and 37b4 (bottom) chromosomes linearized at the dnaA start codon, using the Mauve tool https://darlinglab.org/mauve/mauve.html (accessed on 22 December 2022). Colored blocks indicate locally colinear blocks of homologous sequences with connecting lines showing the relative positions of the two chromosomes. Relative to the SB1003 reference, the regions of the 37b4 sequence that are encoded on the opposite strand (inversions) are shown as downward-pointing blocks of homology, and blank regions indicate sequences lacking homologs in the other chromosome. Numbers give the distance in nucleotides from the *dnaA* start codon. The upward-pointing black arrow shows the location of the SB1003 CPS biosynthesis gene cluster absent from 37b4.

**Figure 3 genes-14-01124-f003:**
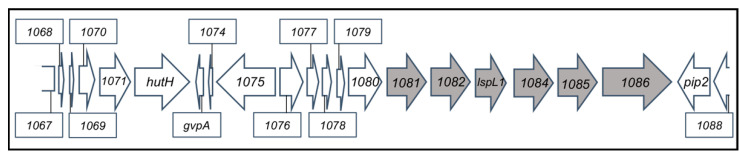
Representation of the WT strain B10 chromosome region that is present in the cosmid pCPS1 that endows strain 37b4 with recipient capability. Numbers give *rcc* locus tags. Orfs shaded in grey encode the proposed CPS biosynthetic genes. The DNA sequence of *rcc01081* to *rcc01088* is identical to that of WT strain SB1003 except for a 3 nt (GGG) deletion in the intergenic region between *lspL1* and *rcc01084* in B10 (unpublished).

**Figure 4 genes-14-01124-f004:**
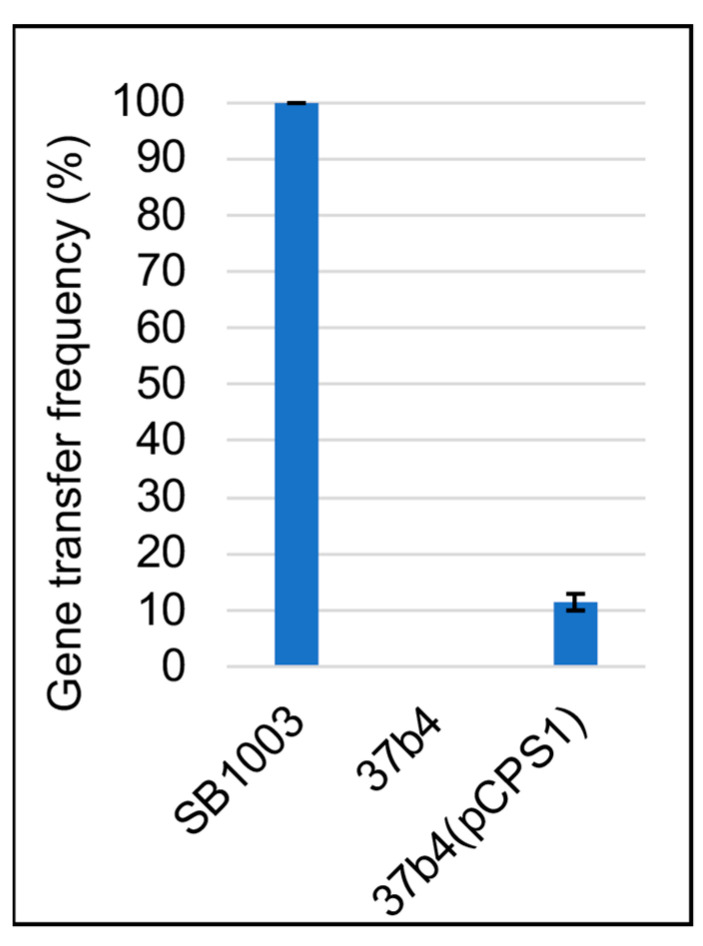
Gene transfer frequencies mediated by RcGTA. The relative number of strains 37b4 and 37b4(pCPS1) kanamycin-resistant colonies due to RcGTA-mediated transfer of locus *rcc02539* that was disrupted by a kanamycin resistance cassette compared to strain SB1003. Blue bars represent the average of two technical replicates, and error bars show the range.

**Figure 5 genes-14-01124-f005:**
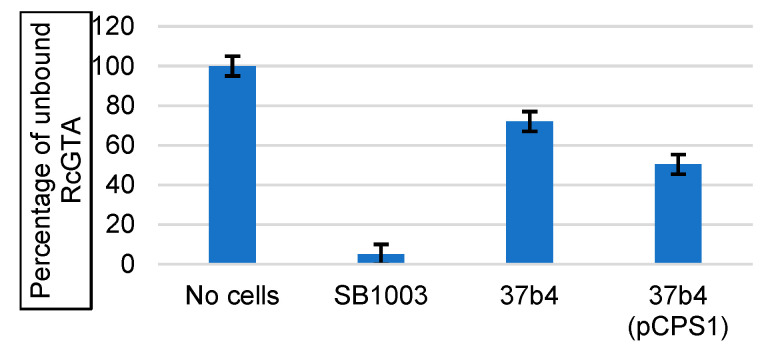
Binding of RcGTA to cells. RcGTA was incubated with SB1003, 37b4, or 37b4(pCPS1) cells to allow binding, cell-bound RcGTA was removed by centrifugation of cells, and residual RcGTA recovered from the supernatant. The relative amount of unbound RcGTA is indicated by the number of kanamycin-resistant SB1003 colonies due to the RcGTA-mediated transfer of locus *rcc02539* disrupted by a kanamycin resistance cassette. Blue bars represent the average of three biological replicates, and error bars show the standard deviation.

**Figure 6 genes-14-01124-f006:**
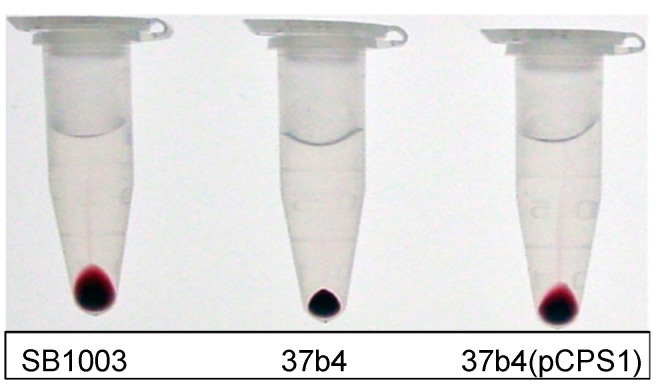
The appearance of cell pellets of strains SB1003, 37b4, and 37b4(pCPS1) after centrifugation of equal numbers of cells.

**Figure 7 genes-14-01124-f007:**
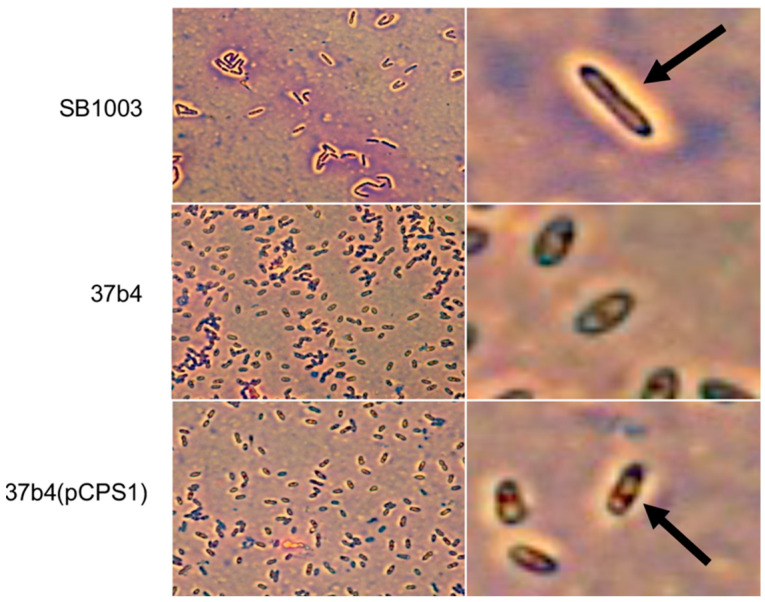
The presence or absence of extracellular capsule around cells of *R. capsulatus* strains (the bright area around cells) viewed in microscopy at 1000× (oil) magnification after being treated with Anthony’s capsule stain. Panels on the (**right**) column show zoomed-in views of corresponding panels on the (**left**). Arrows point to clear zone indicative of a capsule.

**Figure 8 genes-14-01124-f008:**
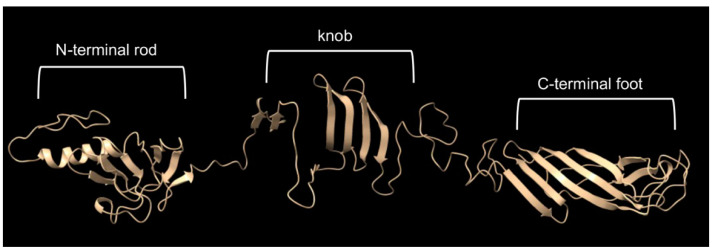
AlphaFold model of the tail fiber monomer.

**Figure 9 genes-14-01124-f009:**
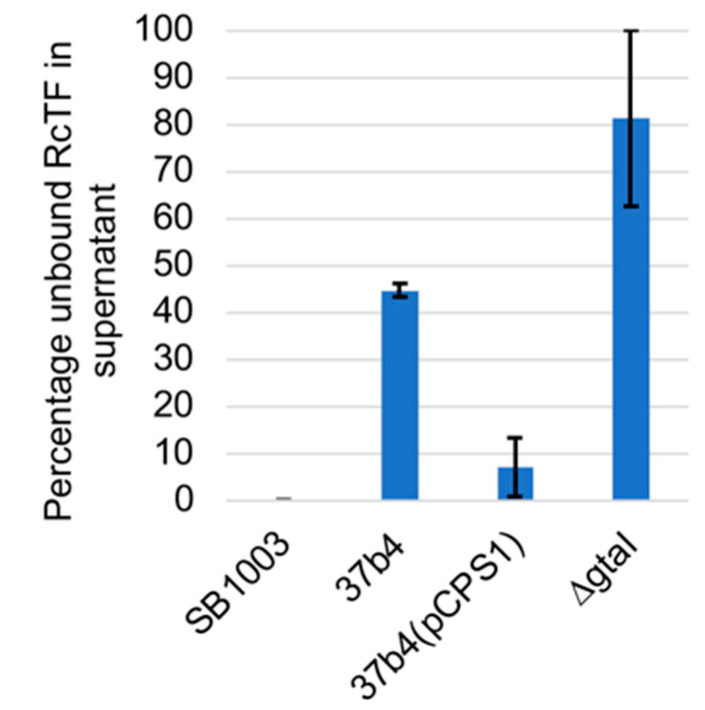
Binding of fluorescent protein-tagged tail fiber to cells of the WT strain SB1003, 37b4, 37b4(pCPS1), and the quorum sensing mutant *ΔgtaI*. Blue bars indicate the fluorescence count of unbound protein molecules in the supernatant subsequent to incubation with excess cells and centrifugation. Higher bars represent weaker binding. Error bars give the standard deviations of biological replicates (*n* = 3).

**Figure 10 genes-14-01124-f010:**
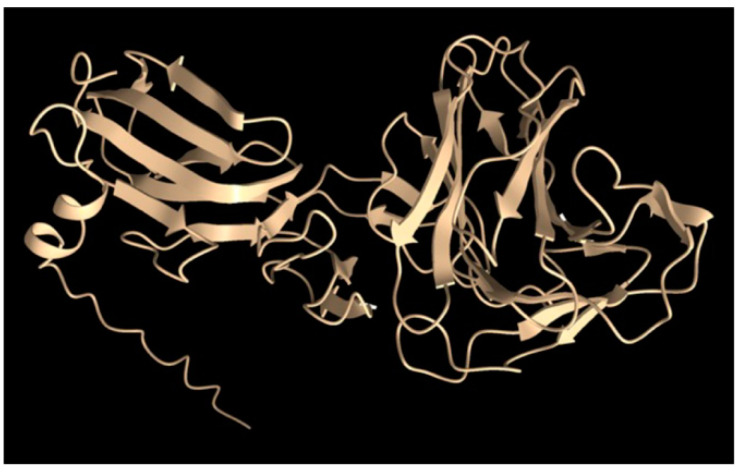
AlphaFold model of the head spike fiber, with the N terminus on the bottom left.

**Figure 11 genes-14-01124-f011:**
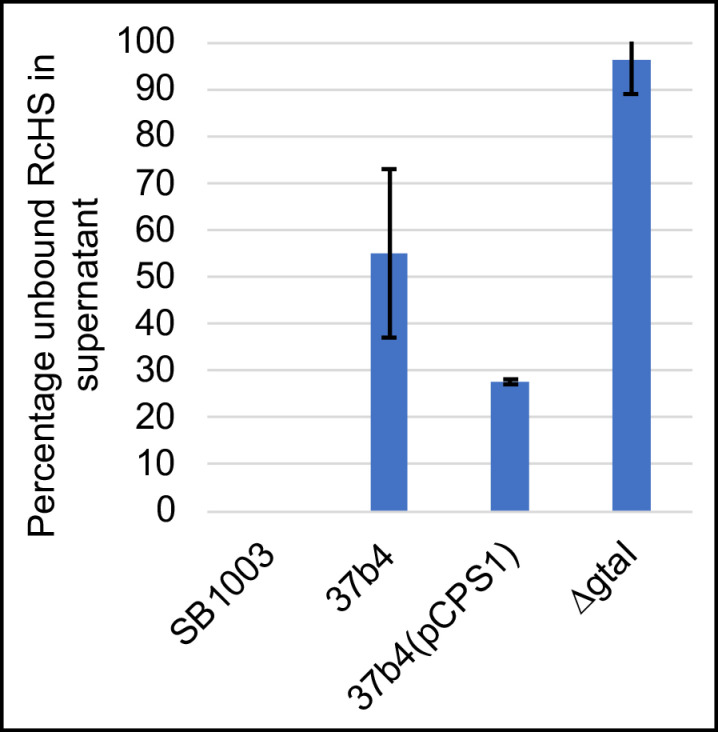
Binding of fluorescent protein-tagged head spike fiber to cells of the WT strain SB1003, 37b4, 37b4(pCPS1), and the quorum sensing mutant *ΔgtaI*. Blue bars indicate the relative fluorescence of unbound protein molecules in the supernatant subsequent to incubation with an excess of cells and centrifugation. Higher bars represent weaker binding. Error bars give the standard deviations of biological replicates (*n* = 3).

**Table 1 genes-14-01124-t001:** Summary of *R. capsulatus* WT strain SB1003 RcGTA receptor gene homologs found in strain 37b4. The E-values of BLASTp and reciprocal blasts are given in Appendix A.

SB1003 Blast Query	Gene Designations	Predicted Function [Reference]	Presence in 37b4
*rcc01081 to rcc01086*	*rcc1083* = *lspL1*	CPS biosynthesis gene cluster [14]	Absent
*rcc01932*	N/A	CPS-related; repeat unit chain initiation [14]	Present
*rcc01958 to rcc01960*	*wzc, wzb, wza*	CPS-related; polysaccharide export [14]	Present
*rcc00197*	*comF*	Competence protein F; required for DNA transport from the periplasm to cytoplasm [41]	Present
*rcc00460*	*comM*	Competence protein M; putative ATPase and helicase-like domains [41]	Present
*rcc00222*	*radC*	DNA repair protein RadC; expressed during competence [41]	Present
*rcc03098*	*dprA*	DNA protecting protein A; transformation-dedicated RecA loader, also involved in competence shut-off [41]	Present
*rcc01751*	*recA*	Recombination protein A; RecA; DNA repair and homologous recombination [42]	Present

**Table 2 genes-14-01124-t002:** Percent mismatch between the strains SB1003 and 37b4 and the corresponding DE442 sequences in regions flanking the insertion site of the kanamycin resistance cartridge in *rcc02539*.

Strain	% Mismatch in 200 bp 5′	% Mismatch in 200 bp 3′	% Mismatch in 400 bp 5′	% Mismatch in 400 bp 3′
SB1003	0	1	0	0.5
37b4	6.5	7	8.2	6.2

## Data Availability

NCBI Bioproject accession number PRJNA933930.

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
