# Peer review of "Extracellular Polysaccharide Receptor and Receptor-Binding Proteins of the Rhodobacter capsulatus Bacteriophage-like Gene Transfer Agent RcGTA"

_genes, 2023, doi:10.3390/genes14051124_

Round 1

Reviewer 1 Report

This article studied how extracellular polysaccharide receptor and receptor-binding proteins of the Rhodobacter capsulatus bacteriophage-like gene transfer agent RcGTA. This article provide a novel contribution of phage-host association study with both computational and experimental efforts. The story  is interesting and we only have two minor concerns about the article .

Q1: As mentioned in Line 84, the RcGTA head spike fiber protein shows great flexibility, how can the AlphaFold structure prediction ensure its representativeness ? Are there any real examples that share a similar structure with the AlphaFold prediction?

Q2: Can you explain the ∆gtaI mutant in section 3.3? It would be even clear to understand if could make explanation of its purpose and significance.

Reviewer 2 Report

The manuscript is well organized and written.  If one plasmid just harboring the target region was transferred to 37b4, the conclusion would be much more solid.  Other Rhodobacter capsulatus strains lacking recipient capability wold be also interesting. The absence of capsular polysaccharide may not be the soly reason for the lack of recipient capability. 

NO further comment. 
